# Vaccine Strategies to Elicit Mucosal Immunity

**DOI:** 10.3390/vaccines12020191

**Published:** 2024-02-13

**Authors:** Yufeng Song, Frances Mehl, Steven L. Zeichner

**Affiliations:** 1Department of Pediatrics, University of Virginia, Charlottesville, VA 22908, USA; bxw6yz@virginia.edu (Y.S.);; 2Department of Microbiology, Immunology, and Cancer Biology, University of Virginia, Charlottesville, VA 22908, USA

**Keywords:** mucosal immunity, mucosal vaccine, mucosal barrier, mucosal adjuvant, vaccine delivery, vaccine development

## Abstract

Vaccines are essential tools to prevent infection and control transmission of infectious diseases that threaten public health. Most infectious agents enter their hosts across mucosal surfaces, which make up key first lines of host defense against pathogens. Mucosal immune responses play critical roles in host immune defense to provide durable and better recall responses. Substantial attention has been focused on developing effective mucosal vaccines to elicit robust localized and systemic immune responses by administration via mucosal routes. Mucosal vaccines that elicit effective immune responses yield protection superior to parenterally delivered vaccines. Beyond their valuable immunogenicity, mucosal vaccines can be less expensive and easier to administer without a need for injection materials and more highly trained personnel. However, developing effective mucosal vaccines faces many challenges, and much effort has been directed at their development. In this article, we review the history of mucosal vaccine development and present an overview of mucosal compartment biology and the roles that mucosal immunity plays in defending against infection, knowledge that has helped inform mucosal vaccine development. We explore new progress in mucosal vaccine design and optimization and novel approaches created to improve the efficacy and safety of mucosal vaccines.

## 1. History of Vaccine Development

Vaccines, one of the most important medical inventions in human history, are essential tools for preventing disease. They have saved countless lives and continue to be a key component of public health efforts, especially in lower-income countries [1].

Historical practices, like variolation, which had their origins in ancient history in the civilizations of East and South Asia, constitute the earliest examples of efforts by humans to prevent infectious disease by prior exposure to infectious agents [2,3,4]. Interestingly, some of these early variolation techniques involved exposing individuals to infectious agents via a mucosal route, such as nasal insufflation of infectious material, pulverized smallpox lesion scabs [3,5]. The public health and military importance of immunization in historical settings is illustrated by the well-known history of General George Washington mandating variolation for troops under his command during the American Revolutionary War [6].

Vaccination by a modified or homolog pathogen began in the late 18th century with Edward Jenner, who found that deliberate infection with cowpox could protect against the more fatal smallpox [7,8,9]. In the 1880s, Louis Pasteur expanded the target of vaccines from smallpox to other pathogens when he discovered that pathogens could be attenuated through repeated passage in environments in which the full complement of pathogenic features was not required for pathogen survival or inactivated by chemical exposure. In 1886, the first inactivated whole-organism vaccines were created against typhoid and cholera [10,11]. The 20th century saw the invention of vaccines that used only parts of the pathogen, including toxoids (1923), pathogen subunits (1944), capsular polysaccharides (1977), protein-conjugated capsular polysaccharides (1987), and recombinant proteins (1986). The use of cell culture to attenuate viruses also became a much-used tool in the 1950s and 1960s [11]. Viral vector-based and mRNA vaccines are some of the more recent vaccine developments of the 21st century [12].

The first vaccines were delivered parenterally using a lancet [9]. Even now, the most common mechanisms of delivery remain the parenteral routes: intramuscular, subcutaneous, or intradermal [13]. Vaccines are evaluated for efficacy through clinical studies, with an important corollary being their ability to provoke a systemic immune response, measured by assessing levels of antibodies against the pathogen [11,14]. While parenterally administered vaccines have made great contributions to human health, and some mucosally-administered vaccines have proved to be safe and effective, such as the live attenuated polio [15,16], rotavirus [17], and influenza [18,19] vaccines and the killed whole-cell cholera vaccine [20,21], most vaccines are not mucosally administered. The advantages of vaccines administered via a mucosal route have led to increasing interest in the development of new mucosal vaccines [13,14,22,23,24]. Routes that have been used for mucosal vaccination include mainly intranasal administration and oral administration, which can deliver vaccine antigens to the GI tract beyond the stomach. Intraoral administration of vaccines, such as sublingual or buccal exposure, has also been used [25,26,27], although these routes have not been as well developed. Rectal vaccine administration has also been studied, and vaccination across urogenital mucosa is also possible, but such routes are generally less appealing due to practical administration and patient acceptability concerns [28].

Parenteral vaccines made of virus-like particles have induced the production of IgA in addition to IgG [29,30,31]. DNA parenteral vaccines have also induced mucosal immunity in animal models [32,33,34]. Some, but not all, adjuvants in parenteral vaccines have promoted mucosal immunity; the mechanism behind this is not yet understood [35]. Nevertheless, most workers in the field consider that an optimal mucosal immune response will result following mucosal exposure to the immunogen [36,37,38,39,40].

## 2. Importance of Mucosal Vaccines and Their Development

Mucosal immunity is of particular importance because mucosal surfaces are the point of entry for more than 90% of human pathogens [41]. The mucosal surfaces represent the first line of the host against infection. Strong mucosal immunity can prevent infection by pathogens across mucosal surfaces and reduce secondary transmission of pathogens shed from a mucosal surface [41,42,43,44]. Even in the event of infection, mucosal immunity at sites of pathogen shedding could prevent forward transmission, lowering infection rates, thereby slowing the emergence of new mutants, reducing morbidity, and blunting potential future pandemics [43,45]. Most vaccines produce a systemic immune response, and while some animal studies have suggested that injected vaccines produce good mucosal immune responses, many parenterally administered vaccines that produce excellent humoral immune responses yield substantially worse mucosal immune responses [35,42]. Several studies have shown administering vaccines at a mucosal surface elicits the most effective mucosal immune responses [36,37,38,39,40].

A mucosal vaccine is a type of vaccine designed to stimulate effective immune responses at the sites of infection, which can help prevent the spread of diseases and is administered through the mucosal surfaces of the host body. Mucosal vaccines have several advantages over traditional vaccines. They can provide more effective protection against diseases by blocking transmission through mucosal surfaces at the site of infection [43,46]. Additionally, most mucosal vaccines can be easier to administer than traditional vaccines, as they do not require needles, syringes, and material to disinfect the injection site [47], which decreases costs and logistical problems. Non-parenteral vaccines can be administered by personnel with less training than is required for parenteral administration [48]. Non-parenteral vaccines may also be more readily accepted by vaccinees who are averse to receiving injections [49,50].

Where mucosal immunity is induced depends on the route of administration. Mucosal vaccines administered across one mucosal surface elicit mucosal immunity where they are administered and in certain other mucosal compartments depending on the administration route. Both nasal and sublingual vaccines can elicit mucosal immunity in the upper and lower respiratory tract and the reproductive tract. Sublingual vaccines also elicit mucosal immunity in the stomach and small intestine. Oral, rectal, and vaginal vaccines induce mucosal immunity in the GI tract, colon and rectum, and reproductive tract [51,52].

The first modern, licensed mucosal vaccine was produced by Albert Sabin, who created an oral polio vaccine with three attenuated strains [10,53]. This oral vaccine induced systemic immunity but, importantly, also local immunity in the GI tract. Replication of the vaccine poliovirus strains also led to secondary exposure to the vaccine strains in non-vaccinated persons, strengthening herd immunity. Mucosal immunity elicited by oral polio vaccines addressed the problem of polio being able to replicate in the gut of people vaccinated with the injectable inactivated vaccine and then infect non-vaccinated individuals [53].

Most of the approved mucosal vaccines employ attenuated pathogens administered orally (Table 1). There have been approximately 30 clinical trials for nasal vaccines, but FluMist^®^ (a live, attenuated influenza vaccine) remains the only one approved in the United States [54]. FluMist^®^ is composed of influenza strains that are temperature-sensitive and cold-adapted, becoming attenuated at body temperature [55,56]. Vaccination with intranasal live, attenuated influenza vaccines yields higher serum IgA and T-cell immunity and decreased viral replication compared to intramuscular inactivated vaccines [57,58]. However, they also have decreased hemagglutination inhibition, microneutralization, and IgGs [57,59].

Both parenteral and mucosal vaccines may promote cellular immunity; however, only mucosal vaccines promote mucosal tissue-resident memory T-cells and B-cells. Therefore, mucosal vaccines enable a more direct and immediate response to an invading pathogen [59,60,61]. This effect for viral vaccines depends on the pathogen being live or in another viral vector; an inactivated virus administered intranasally does not induce cellular immunity [59,60], although it may be possible to develop new strategies to enhance the production of cell-mediated immunity by non-living vaccines with adjuvants, vehicles, or novel dosing schedules. One study has shown that inactivated influenza virus administered intranasally induced a Th1, Th2, and Th17 response, whereas the same vaccine administered intramuscularly only induced Th1 and Th2 [62].

Killed whole-cell cholera vaccines (Dukoral^®^, ShanChol™, and Euvichol-Plus^®^, available outside the USA) are mucosal vaccines approved by the WHO, using an inactivated whole organism [63,64,65]. VAXCHORA^®^, a live attenuated bacterial vaccine, is another FDA-approved single-dose oral vaccine for people between the ages of 2 and 64 traveling to areas of active cholera transmission [66,67,68]. There is research into other live attenuated oral bacterial vaccines for tuberculosis and enterotoxigenic *E. coli* and a live attenuated intranasal vaccine for pertussis [69,70,71,72,73,74].

Although much effort has been directed at the development of mucosal vaccines, significant challenges remain. The number of approved mucosal vaccines for humans is low compared to the number of pathogens that enter a mucosal surface. Only 8 out of 96 licensed vaccines in the United States are mucosal (https://www.fda.gov/vaccines-blood-biologics/vaccines/vaccines-licensed-use-united-states (accessed on 21 January 2024)).

**Table 1 vaccines-12-00191-t001:** List of FDA and WHO-approved mucosal vaccines for human use [75,76,77].

**FDA-Approved Vaccines**						
**Pathogen**	**Product Name**	**Trade Name**	**Manufacturer**	**Delivery Route (Form)**	**Formulation**	**Initial Approval**
Adenovirus Types 4 and 7	Adenovirus Type 4 and Type 7 Vaccine, Live, Oral	Adenovirus Type 4 and Type 7 Vaccine, Live, Oral	Barr Labs, Inc. North Wales, PA, USA	Oral (tablets)	Live attenuated	2011 (military use only)
Influenza Type A (H1N1)	Influenza Vaccine, Live, Intranasal (Monovalent, Type A H1N1)	Influenza A (H1N1) 2009 Monovalent Vaccine Live, Intranasal	MedImmune, LLC Gaithersburg, MD, USA	Intranasal (spray)	Live attenuated	2003
Influenza Types A and B	Influenza Vaccine, Live, Intranasal (Trivalent, Types A and B)	FluMist	MedImmune, LLC Gaithersburg, MD, USA	Intranasal (spray)	Live attenuated	2003
Influenza Types A and B	Influenza Vaccine, Live, Intranasal (Quadrivalent, Types A and B)	FluMist Quadrivalent	MedImmune, LLC Gaithersburg, MD, USA	Intranasal (spray)	Live attenuated	2003
Rotavirus	Rotavirus Vaccine, Live, Oral	Rotarix	GlaxoSmithKline Biologicals Rixensart, Belgium	Oral (liquid)	Live attenuated	2008
Rotavirus	Rotavirus Vaccine, Live, Oral, Pentavalent	RotaTeq	Merck Sharp & Dohme Corp Whitehouse Station, NJ, USA	Oral (liquid)	Live attenuated	2006
*Salmonella typhi*	Typhoid Vaccine, Live, Oral, Ty21a	Vivotif	Berna Biotech, Ltd. Berne, Switzerland	Oral (capsule)	Live attenuated	1989
*Vibrio cholerae*	Cholera Vaccine, Live, Oral	Vaxchora	Emergent Travel Health, Redwood City, CA, USA	Oral (liquid)	Live attenuated	2016
**WHO (but not FDA)-Approved Vaccines**						
**Pathogen**	**Product Name**	**Trade Name**	**Manufacturer**	**Delivery Route (Form)**	**Formulation**	**Initial Approval**
Influenza Type A (H1N1)	Influenza Vaccine, Live, Oral, Pandemic	Nasovac	Serum Institute of India Pvt. Ltd., Hadapsar, India	Intranasal (spray)	Live attenuated	2012
Influenza Type A (H5N1)	Influenza Vaccine, Live, Pandemic	Pandemic Live Attenuated Vaccine	AstraZeneca Pharmaceuticals LP., Nijmegen, Netherlands	Intranasal (spray)	Live attenuated	2020
Influenza Types A and B	Influenza Vaccine, Live, Oral, Seasonal	Nasovac-S	Serum Institute of India Pvt. Ltd., Hadapsar, India	Intranasal (spray)	Live attenuated	2015
Poliovirus	Polio Vaccine—Oral (OPV) Bivalent Types 1 and 3	Biopolio B1/3	Bharat Biotech International Limited, Hyderabad, India	Oral (liquid)	Live attenuated	2017
		Bivalent OPV Type 1 and 3 Poliomyelitis Vaccine, Live (Oral)	Panacea Biotec Ltd., Malpur, India	Oral (liquid)	Live attenuated	2018
		Bivalent Oral Poliomyelitis Vaccine Type 1 & 3 (bOPV 1 & 3)	PT Bio Farma (Persero), Bandung, Indonesia	Oral (liquid)	Live attenuated	2010
		Bivalent type 1 & 3 Oral Poliomyelitis vaccine, IP (bOPV1 & 3)	Haffkine Bio Pharmaceutical Corporation Ltd., Mumbai, India	Oral (liquid)	Live attenuated	2010
		Oral Bivalent Types 1 and 3 Poliomyelitis Vaccine	Sanofi Pasteur, Lyon, France	Oral (liquid)	Live attenuated	2011
		Polio Sabin One and Three	GlaxoSmithKline Biologicals SA, Rixensart, Belgium	Oral (liquid)	Live attenuated	2009
		Poliomyelitis Vaccine (live, oral attenuated, human Diploid Cell), type 1 and 3	Beijing Institute of Biological Products Co., Ltd., Beijing, China	Oral (liquid)	Live attenuated	2017
		Poliomyelitis Vaccine (Oral), Bivalent types 1 and 3	Serum Institute of India Pvt. Ltd., Hadapsar, India	Oral (liquid)	Live attenuated	2013
Poliovirus	Polio Vaccine—Oral (OPV) Monovalent Type 1	Monovalent Oral Poliomyelitis Vaccine Type 1 (mOPV1)	PT Bio Farma (Persero), Bandung, Indonesia	Oral (liquid)	Live attenuated	2009
		Monovalent type 1 Oral Poliomyelitis vaccine, IP (mOPV1)	Haffkine Bio Pharmaceutical Corporation Ltd., Mumbai, India	Oral (liquid)	Live attenuated	2009
		Polio Sabin Mono T1	GlaxoSmithKline Biologicals SA, Rixensart, Belgium	Oral (liquid)	Live attenuated	2009
Poliovirus	Polio Vaccine—Oral (OPV) Monovalent Type 2	Monovalent Oral Poliomyelitis Vaccine Type 2	PT Bio Farma (Persero), Bandung, Indonesia	Oral (liquid)	Live attenuated	2019
		ORAL MONOVALENT TYPE 2 POLIOMYELITIS VACCINE (mOPV2)	Sanofi Pasteur, Lyon, France	Oral (liquid)	Live attenuated	2016
		Polio Sabin Mono Two (oral)	GlaxoSmithKline Biologicals SA, Rixensart, Belgium	Oral (liquid)	Live attenuated	2011
Poliovirus	Polio Vaccine—Oral (OPV) Monovalent Type 3	Polio Sabin Mono Three (oral)	GlaxoSmithKline Biologicals SA, Rixensart, Belgium	Oral (liquid)	Live attenuated	2010
Poliovirus	Polio Vaccine—Oral (OPV) Trivalent	BIOPOLIO	Bharat Biotech International Limited, Hadapsar, India	Oral (liquid)	Live attenuated	2015
		Oral Poliomyelitis Vaccines (Oral Drops)	PT Bio Farma (Persero), Bandung, Indonesia	Oral (liquid)	Live attenuated	2020
		Poliomyelitis vaccine (Oral) IP	Haffkine Bio Pharmaceutical Corporation Ltd., Mumbai, India	Oral (liquid)	Live attenuated	2006
Rotavirus	Rotavirus Vaccine, Live, Oral	Rotasil	Serum Institute of India Pvt. Ltd., Hadapsar, India	Oral (liquid)	Live attenuated	2018
		Rotasil Thermo	Serum Institute of India Pvt. Ltd., Hadapsar, India	Oral (liquid)	Live attenuated	2020
		Rotavac	Bharat Biotech International Limited, Hyderabad, India	Oral (liquid)	Live attenuated	2018
		Rotovac 5D	Bharat Biotech International Limited, Hyderabad, India	Oral (liquid)	Live attenuated	2021
*Vibrio cholerae*	Cholera Vaccine, Inactivated, Oral	Dukoral	Valneva Sweden AB, Stockholm, Sweden	Oral (liquid)	Inactivated (with recombinant cholera toxin subunit B)	2001
		Euvichol	EuBiologics Co., Ltd., Chuncheon-si, South Korea	Oral (liquid)	Inactivated	2015
		ShanChol	Sanofi Healthcare India Private Limited, Medchal, India	Oral (liquid)	Inactivated	2011

The veterinary vaccine industry has many more licensed mucosal vaccines, targeting a variety of pathogens and given to many different types of animals, including cattle, poultry, and fish. In addition to the immunological benefits, these vaccines are cheaper and easier to administer to large groups than parenteral vaccines [78]. Many swine and some poultry vaccines can be administered in drinking water, including Coliprotec F4, which consists of avirulent live *E. coli* and protects against post-weaning diarrhea in swine [79]. Live attenuated *Salmonella* poultry vaccines can be administered to flocks using a large sprayer [80,81,82]. While most mucosal vaccines are live, the rabies vaccine for control of rabies in wildlife uses a viral vector to express the rabies glycoprotein and is administered orally in bait [83,84].

In addition to a need for mucosal vaccines against more pathogens, there is also a need to develop new, more effective, less expensive mucosal vaccine platforms. Many of the vaccine platforms invented in the last hundred years and used for parenteral vaccines have not yet been successfully applied to mucosal vaccines or only applied in limited ways with minimal successful dissemination worldwide. For example, the intranasal hepatitis B vaccine, HeberNasvac^®^, which uses recombinant virus-like particles, represents one successful application, but it is only approved in Cuba [54].

There are many new platforms in the research phase for mucosal vaccines. Some include viral vectors, which can mimic natural infections and elicit both an innate and adaptive immune response. At least one viral-vector vaccine for SARS-CoV-2 is in the human clinical trial stage, and a viral-vector vaccine for tuberculosis has been tested in mice [85,86,87]. A non-viral vector (N1,N3-dicarbamimidoyl-5-methylisophthalamide (BGG)) carrying antigen-expressing plasmid DNA has also been developed and used in an intranasal SARS-CoV-2 vaccine [88,89]. Oral mRNA vaccines are also being developed [90]. Several intranasal vaccines that combine protein subunits and adjuvants are being designed and tested against pertussis, tuberculosis, and SARS-CoV-2 [91,92,93,94,95,96]. Outer membrane vesicles, which have previously been used in parenteral meningitis B vaccines, are now being used for intranasal pertussis and SARS-CoV-2 vaccines [97,98,99]. Recent progress on improved vaccines against pertussis and tuberculosis via nasal delivery demonstrates that they elicit more effective immunity compared with traditional vaccine formulations that have been used for decades [100,101]. Broad spectrum antigen-specific antibody responses and cellular-mediated immunity are achieved after those mucosal vaccinations in clinical trials, even with a single dose [101]. Sustained protective immunity up to 12 months after vaccination [101] and sterilizing immunity was observed in animal models [100].

Additionally, our lab is developing a vaccine platform that uses killed whole-cell genome-reduced bacteria, that uses an autotransporter to express the antigen of interest and immune modulators that can be used non-parenterally [102].

## 3. The Mucosal Barrier

The body’s mucosal surfaces have an enormous area, about 200 times greater than the skin. Mucosae line many important organs and have been classified into two different types (Figure 1). Type I mucosa are found in the respiratory tract, most of the gastrointestinal tract, tonsils, and adenoids, whereas type II mucosa includes the oral cavity and urogenital tract [42,43,103]. The mucosal surface is where a pathogen present in the environment first encounters and then enters the host [104]. The mucosal barrier, in most organs, is just a single layer of epithelial cells separating the environment from the interior, serves as both a barrier to pathogens and other potentially harmful factors in the environment and the interior of the organism, and in some organs, a surface that exchanges essential nutrients and gases. The mucosal barrier also, in some organs, like the respiratory and GI tracts, enables the immune system to sense and sample the environment for antigens and transmit those antigens on to the immune system to initiate a response.

Most mucosal tissue consists of three layers: epithelium, lamina propria, and muscularis mucosae. The mucosal epithelium lines the inner surfaces of organs and body cavities that are exposed to the external environment. The epithelium is composed of cells joined by tight junctions and faces a complex environment rich in microorganisms [42]. Epithelial cells protect the host from irritants and toxins by secreting a thick, gel-like mucus and arrange themselves in different ways to accomplish different functions, such as epithelia having one or several layers of cells [105]. Epithelial cells are one of the main types of cells in the human body. They are found in the lining of organs and body cavities, and they also form glands. Epithelial cells have many functions, including secretion, absorption, sensation, protection, and transport [106,107]. Mucosal epithelium can be defined as the following different types [108] (Figure 2): (1) Nonkeratinized epithelium, which is found in the oral cavity, esophagus, vagina, and anus is composed of several layers of cells, with the outermost layer being moist and alive. This type of epithelium mostly provides protection against mechanical stress and abrasion [107,108,109]; (2) columnar epithelium is a type of epithelium found in the stomach, intestines, respiratory tract except for alveoli, and gallbladder. It is composed of tall, narrow cells that are tightly packed together and are mainly specialized for absorption and secretion [107,108,109]; (3) squamous epithelium is a type of epithelium found in the lungs, blood vessels, and body cavities. It is composed of flat, scale-like cells that are tightly packed together. This type of epithelium serves as a barrier against infection and injury [107,108,109]. In the lungs, in the alveoli, gas exchange occurs across thin, Type I squamous epithelial cells, while cuboidal Type II cells secrete surfactant. In other parts of the respiratory tract, the epithelia include ciliated cells, nonciliated columnar cells, mucous (goblet) cells, brush cells, basal cells, and associated submucosal cells. Ciliated cells play critical roles in mucociliary clearance to propel pathogens and inhaled particles trapped in the mucous layer out of airways by beating in metachronal waves of cilia [110]. The classification of epithelial cells into different types according to their shapes and cell layer numbers reflects their physiological and host defense functions. Simple squamous cells are thin, flat cells that line the blood vessels, lungs, and body cavities. They are involved in the exchange of gases and nutrients [107,108]. Simple cuboidal cells are cube-shaped cells that line the kidney tubules and the ducts of glands. They are involved in secretion and absorption [106,108]. Simple columnar cells are elongated cells that line the digestive tract and the uterus. Their major functions are secretion, absorption, and protection [111]. Stratified squamous cells are flat cells that form the outer layer of the skin and the lining of the mouth, throat, and vagina. They are mainly involved in physical protection [112]. Stratified cuboidal cells are cube-shaped cells that form the lining of the sweat glands and the salivary glands. Their functions are mostly in secretion [108]. Stratified columnar cells are elongated cells that form the lining of the male urethra and the ducts of some glands [107]. They are involved in secretion and protection. Pseudostratified columnar cells are elongated cells that line the respiratory tract. They are involved in secretion and protection.

The epithelial layer is the site of immunological activities and constitutes both a physical and biochemical barrier that enables the transfer of nutrients, antigens, and other immune factors while resisting the transmission of pathogens and toxins [42,43,105]. Epithelia, together with associated glands, provide innate non-specific defenses via the production and secretion of materials such as mucins, antimicrobial proteins, enzymes, secretory antibodies, buffers, prebiotics for microbiome modulation, salts, water, hormones, and other bioactive molecules, and other components of the intraluminal mucosal environment [42,113,114]. The epithelium of many organ systems also has important neural sensing, including temperature and molecular sensors for taste and smell and neuroendocrine functions [115,116,117,118].

Epithelial cells replace themselves frequently; they have a high cell turnover rate, which helps to protect against pathogen invasion. Some epithelia include cells with cilia, which move mucus, and small particles trapped in mucus, including pathogens. Besides their physical barrier function, epithelial cells play active and important roles in mucosal immune defense. They can function in immune surveillance by sensing dangerous pathogen components through pattern-recognition receptors (PRRs), such as Toll-like receptors (TLRs), to stimulate innate and adaptive immune responses by secreting cytokine and chemokine signals as a result of cell activation following receptor binding, to signal underlying mucosal immune cells, like dendritic cells (DCs) and macrophages [42,43,109,119]. Some epithelial cells have specialized functions in immune sensing. For example, M cells are found in the epithelium of the gut-associated lymphoid tissues (GALT) and are involved in the uptake and transport of antigens from the lumen of the gut to the underlying immune cells [120]. Similarly, analogous cells in the respiratory epithelium are involved in the uptake and transport of antigens from the airway lumen to the underlying immune cells [121]. In addition, other components of the epithelium, like tuft cells, are found in the respiratory and gastrointestinal epithelium and are involved in the detection of parasitic infections [122,123] as well as intraepithelial lymphocytes (IELs) that function in immune surveillance [124,125].

The middle layer of the mucosa is called lamina propria, which is a loose connective tissue attached to the epithelium and composed of connective tissue, nerves, blood and lymphatic vessels, and associated lymphoid tissue. The major functions of the lamina propria include a structural role, supplying blood to the epithelium, hosting lymphoid tissues, and binding epithelial structures to smooth muscle [113,126]. Lamina propria neural components work together with smooth and striated muscle to help modulate the conformation of the epithelium [127].

The muscularis mucosae, a layer of smooth muscle, is the deepest layer of the mucosa. It varies in thickness throughout the digestive, respiratory, and genitourinary tracts and, in humans, is most active in the stomach and GI tract. The muscularis mucosae provides a motor function that keeps the mucosa in motion, helping the mucosa to stretch and contract along with the organs [113,126]. Impairment of the mucosal barrier can lead to impaired immune functions and responses as well as inflammation [128,129].

## 4. Mucosal Immunity

Mucosal immunity is the immune response that occurs at the mucosal surfaces of the body. The mucosal immune system constitutes a complex network of innate immune components, immune cells, and antibodies working together to maintain the integrity of the mucosal barrier and mediate immune responses to pathogens (Figure 1) [43,45,103]. An ideal mucosal vaccine should elicit durable, effective local immunity. Detailed knowledge of the mucosal immune system can help inform the development of new mucosal vaccines.

Innate immune responses activate the adaptive immune system, which is composed of T and B lymphocytes that can recognize specific antigens from different pathogen-derived molecules. The mucosal cellular immune response begins with antigen-presenting cells (APC). DC cells are classical APCs. They process captured antigens and, after activation and maturation, present antigens to naïve T-cells, yielding clonally expanded and differentiated Th cell subsets including Th1, Th2, Th17, and Treg cells. Each cell type subset has cell type-specific cytokine secretion profiles [104,130]. Once activated, the lymphocytes proliferate and differentiate into effector cells that can eliminate the pathogen via direct cell killing by T-cells and/or immunoglobulins, aided by secretion of cytokines and chemokines that promote inflammation and serve as chemoattractants for additional lymphocytes, monocyte/macrophages, neutrophils, and in some circumstances, eosinophils and basophils [103,131].

The mucosal immune system is comprised of anatomic and physiologic components with distinct functions, which can vary in the mucosa of different tissues [132,133]. Based on functional and anatomical properties, mucosal tissues can be divided into inductive and effective sites [132]. The mucosal defense mechanisms can be defined as physical barriers (epithelial lining, mucus, cilia function) and biochemical factors (pH, antimicrobial peptides) [134]. Antimicrobial peptides (AMPs) are a class of chemical factors produced by epithelial cells and immune cells, often at increased levels in response to microbial invasion [135,136,137,138]. They are small peptides with broad-spectrum antimicrobial activity against infections. Mechanisms of action of AMPs include disrupting the microbial cell membrane, inhibition of protein synthesis, and inducing an oxidative stress response [135]. AMPs are an important component of the non-adaptive immune system because they provide immediate protection against pathogens before the adaptive immune system can mount a specific response [138]. In addition to their antimicrobial activity, AMPs also have immunomodulatory functions, such as promoting wound healing, regulating inflammation, and enhancing the recruitment of immune cells [136].

In addition to the AMPs, the microbiome of mucosal compartments is another component of the mucosal immune system, consisting of a complex ecosystem of microorganisms that inhabit the mucosal surfaces of the body and lumens of body cavities [139,140,141]. The microbiome plays an important role in maintaining the health of the host by regulating the immune system, protecting against pathogens, and aiding in digestion [139]. The composition of the microbiome varies depending on the location of the mucosal sites. The microbiome of the respiratory tract is dominated by bacteria such as *Streptococcus*, *Haemophilus*, and *Moraxella* [140], while the microbiome of the gastrointestinal tract is more diverse and includes bacteria such as Bacteroidaceae, Firmicutes, and Actinomycetota [140]. The microbiome of mucosal compartments can interact with the host immune system by stimulating the production of antimicrobial peptides to protect against pathogens and regulating the development and function of immune cells, T-cells, and B-cells [142]. The microbiome of some mucosal compartments can help defend against invading pathogens by altering the mucosal environment to make it more hostile to invading pathogens, occupying ecological niches to exclude invaders, and producing antimicrobial substances, and the host can work to modulate the microbiome to help accomplish this. An interesting example is the female genital tract. A healthy vagina has a low pH, promoted by the production of organic acids produced by *Lactobacillae*, the dominant component of a healthy vaginal microbiome. The vaginal epithelium secretes glycogen, which helps “feed” the lactobacilli, so the vaginal epithelium helps defend against potential pathogens by providing endogenous prebiotics that help modulate the microbiome to provide a more effective defense [143,144,145,146,147,148].

Cytokines secreted by mucosal cells play crucial roles in controlling the growth and activity of other immune cells by establishing cell–cell interactions and facilitating cellular signaling. The tightly regulated cytokine signaling network regulates both innate and adaptive immune responses across mucosal barriers [149,150].

The mucosal effector sites are comprised of the lamina propria regions in mucosal barriers of the upper respiratory, gastrointestinal, and reproductive tracts, along with secretory glandular tissues. There are several different types of antigen-specific mucosal effector cells in effector sites, including T helper cells (TH), regulatory T-cells (Tregs), and IgA-producing plasma cells, which, together, compose the humoral and cellular immune responses localized in mucosal sites [132,133,151]. Humoral immune responses are mediated by immunoglobulin molecules produced by plasma cells [152]. Although some IgG is excreted into mucosal cavities, IgA is the predominant immunoglobulin isotype in mucosal immunity and widely exists in mucosa in the respiratory, gastrointestinal, and urogenital tracts. It is also present in tears, saliva, and colostrum [153,154]. IgA in sera is found as monomers. In contrast, mucosal IgA is polymeric [153,155]. The induction of SIgA, secretory IgA, serves as the first line of defense to protect the mucosal surface from dangerous substances such as toxins and microorganisms by preventing their access and entry through epithelial receptors. Major IgA production sites in the mucosa are gut-associated lymphoid tissues (GALT), including Peyer’s patches (PPs), isolated lymphoid follicles, and mesenteric lymph nodes (MLNs), producing more than 90% of the SIgA made by the body [153,156]. PPs are the principal precursor source of IgA-producing plasma cells [157], secreting an average of 3 g of SIgAs into the gut lumen of an adult human each day [158]. Antigens are taken up by mucosal immune cells such as dendritic cells (DCs) [159], follicle-associated epithelium containing microfold (M) cells [160], follicle-associated epithelium (FAE) [161], and small intestine goblet cells (GCs) [162]. Upon antigen processing and presentation, APCs, T-cells, and B-cells are activated, and IgA class switch recombination and somatic hypermutation (SHM) are later initiated in the mucosal B-cells [163,164] in which activation of SMAD3/4 and Runx3 downstream of the TGF-β recognition signaling pathway also contributes to IgA production [154,164,165]. The mechanisms of IgA class switch recombination are divided into T-cell-dependent (TD), which requires the interaction of CD40 on B-cells with CD40L, derived from T-cells, resulting in high-affinity antigen-specific neutralizing IgA production [166], and T-cell-independent (TI) which produce commensal-reactive IgA through stimulation of B-cell activating factor of the TNF family (BAFF) and A proliferation-inducing ligand (APRIL) secreted from innate immune cells such as innate lymphoid cells (ILCs) and plasmacytoid dendritic cells (pDCs) [166,167,168,169]. Human B-cells produce two subclass of IgA molecules, IgA1 and IgA2, with similar receptor binding profiles but different topographies [164]. IgA1 exists in both systemic and mucosal districts, whereas IgA2 is mostly present in mucosal districts colonized by plentiful microbiota, including the mucosa of the distal intestinal tract and the urogenital tract [164].

B-cell activation occurs at multiple mucosal sites, promoting B-cell differentiation and maturation and production of antigen-specific IgG antibodies as well as secretory IgA (SIgA), which can neutralize the pathogens and prevent invasion through the mucosal surface by antibody neutralization activities or Fc-mediated functions including Antibody-Dependent Cellular Cytotoxicity (ADCC), Antibody-Dependent Cellular Phagocytosis (ADCP), and Antibody-Dependent Cell-mediated Viral Inhibition (ADCVI) [103,170,171]. IgG excretion is mediated by the neonatal Fc receptor (FcRn) [172], whereas IgA molecules are secreted via polymeric Ig receptor (pIgR), which is an epithelial membrane receptor that interacts with and transcytoses J chain-containing dimeric IgA (and pentameric IgM) produced by lamina propria plasma cells [173]. Most of the antibodies produced in the host take place locally in the mucosal surfaces where the predominant form is the dimeric IgA by pIgR-mediated export, which demonstrates the importance of IgA in mucosal immunity as most pathogens are encountered by the host mucosal surfaces [151,173]. Besides pathogen clearance via ‘immune exclusion’, SIgA also has other functions like reinforcement of the epithelial barrier, facilitation of APC to capture antigens by promoting downstream local immune responses, and excludes the pathogenic microorganisms from the epithelial surface by forming a biofilm of non-pathogenic microorganisms as a result of anchoring within the mucus [151,174].

IgG is the most abundant antibody in blood and is involved in the systemic immune response. IgG can also be found in type I mucosae, such as the respiratory and gastrointestinal tracts; specific IgG is increased in response to local infections or vaccination [154,156,175,176,177]. IgA is mostly found in type I mucosae like the upper respiratory tract, whereas the IgG is the predominant protective isotype in type II mucosae, such as corneal, oral, esophageal, lower respiratory tract, and lower female reproductive tracts, due to the expression of FcRn but not pIgR [175]. The production of IgG in mucosal immune responses is regulated by a complex interplay between local immune cells and the systemic immune system [176]. B-cells in mucosal tissues can be activated by APCs. Activated B-cells can then differentiate into plasma cells that produce IgG [177]. The production of IgG in mucosal tissues can also be influenced by cytokines and chemokines produced by immune cells in response to infection or inflammation [121,177,178]. In addition, some IgM could be found in respiratory and urogenital tract mucosa, whereas IgD constitutes a significant fraction of antibodies in the upper segments of human respiratory and digestive tracts [154].

The mucosal immune sensing sites are comprised of mucosa-associated lymphoid tissue (MALT), including the gut-associated lymphoid tissues (GALT), nasopharyngeal-associated lymphoid tissue (NALT), and lymphoid sites. MALT is composed mostly of dendritic cells, macrophages, innate lymphoid cells, mucosal-associated invariant T-cells, intraepithelial T-cells, regulatory T-cells (Treg), as well as IgA secreting plasma cells and a source of memory B- and T-cells that migrate to effector sites to elicit immune responses [103,132].

A typical pathogen invasion begins from exposure through routes such as inhalation, ingestion, or direct contact with a mucosal surface, such as sexual intercourse. Then, pathogens attach to the epithelium and penetrate through the mucosal layer to enter the host either by infecting epithelial cells or by gaining access to cells in the submucosa through a microabrasion in the mucosal barrier, or transport across a mucosal cell or through the actions of a mucosal immune cell, such as dendritic cells, followed by establishment of target cell infection in the lamina propria [175]. The immune system responds to pathogens in the mucosa by initiating a series of immune responses, which are composed of a variety of immune cells, including T-cells, B-cells, macrophages, dendritic cells (DCs), natural killer (NK) cells, innate lymphoid cells (ILCs), and plasma cells. When the mucosa encounters a pathogen, the pathogen is first recognized by the innate immune system, which is composed of antigen-presenting immune cells through PRRs. These innate immune cells can engulf and destroy pathogens or present them to other immune cells for further processing [179]. The follicle-associated epithelium containing microfold (M) cells, one type of lymphoid follicle cell found in MALT, are responsible for antigen capture and uptake in the local lumen sites, followed by antigen presentation to APCs like DCs and macrophages, where the antigens are further processed and presented to lymphocytes [179,180,181]. Th cells play critical roles in mucosal immune responses, such as the Th17 cell, a Th subset that mostly secretes interleukin-17 (IL-17), which has been reported to help promote protective mucosal immunity induced by vaccines with increased SIgA levels by upregulating the polymeric Ig receptor expression levels in mucosal epithelial cell surfaces and promote T-cell-independent B-cell maturation and differentiation [103,104].

After mucosal vaccination or pathogen infection, a subset of memory lymphocytes, known as tissue-resident memory cells, are preferentially elicited, while little to no numbers of those cells were observed following parenteral immunizations [182]. Those cells, maintained within non-lymphoid barrier tissues, act as sentinels, providing rapid local recall of immune responses to previously encountered antigens, driving accelerated pathogen clearance via mechanisms that involve enhanced cytokine expression and functional effector molecules secretion [61]. Reactivation of long-lived tissue-resident memory lymphocytes can trigger enhanced innate and adaptive immune responses in the host as a whole, beyond the mucosal compartment, mediated by cytokines [61].

When local immune responses are activated after mucosal vaccination, antigens are taken up and processed by APCs, like mucosal epithelial cells and DCs [47]. Then, immune cells are polarized by the antigen-laden APCs, and those activated immune cells migrate via lymphatic vessels to draining lymph nodes, which promotes a systemic immune response, leading to further T- and B-cell proliferation, differentiation, and maturation. Activated T- and B-cells then circulate through the bloodstream, establishing systemic immunity [42]. During the crosstalk between mucosal and systemic immunity, activated lymphocytes express specific homing receptors that guide them in their migrations between mucosal and systemic sites. This creates a feedback loop where systemic immune cells contribute to local mucosal immunity and vice versa [42,47]. Cytokines produced by both local and systemic immune cells also play a crucial role in coordinating the overall immune response, such as IL-17 produced by Th17 cells, promoting both local mucosal inflammation and systemic antibody production [42,47].

Immunological memory was long defined as a characteristic only belonging to adaptive immunity. This traditional thinking has been challenged recently as evidence has been developed showing that, after infection or vaccination, some prototypical innate immune cells, such as myeloid cells or natural killer cells, can also display long-term nonspecific heterologous protective immunity when encountering a second stimulation [101,183,184]. This previously unrecognized property of immune phenomena of hyper-reactive types of innate immune memory, which leads to an enhanced innate immune protective outcome, has been termed “trained innate immunity” (TII [183]). Mechanistic studies have demonstrated epigenetic, metabolic, and functional reprogramming of myeloid cells and myeloid progenitors in the bone marrow and peripheral tissues, such as gut and lung mucosa, that mediate a persistent change in the homeostatic state of the innate immune cells that endure long after the initial antigenic or microbial exposure/clearance, with increased responsiveness and production of inflammatory mediators upon secondary stimulation, as well as enhanced capacity to eliminate infection. This yields an enhanced innate host defense against the same or unrelated pathogens [183,185]. TII is now recognized as a potential correlation of protection against M. tuberculosis infection, and studies also reveal that the respiratory mucosal route of TB vaccination-induced robust memory airway macrophages in the lung provides a type of TII capable of potent protection against M. tuberculosis in the early stage of infection, independent of T-cell immunity [101,186]. Thus, the concept of TII has received renewed attention not only in light of new knowledge with regard to innate immune memory but also in aiding in designing new generations of vaccines that combine classical immunological memory and TII.

## 5. Mucosal Vaccine Formulation and Delivery Strategies

Knowledge of the mucosal immune system can inform mucosal vaccine design, enhancing effective humoral immune responses mediated by IgA and IgG through the promotion of optimal Th cell responses with maximal B-cell stimulation, more profound CTL responses by incorporation of well-defined protection-correlated antigen-specific T-cell epitopes in the context of Human leukocyte antigens (HLA) alleles, incorporation of adjuvants that can augment innate immune responses by APCs to elicit enhanced and long-lasting protective immunity, and technologies like nanoparticles, liposomes, or virus-like particles to improve the antigen delivery efficacy to mucosa [23,187].

Several currently approved mucosal vaccines are mostly based on traditional live attenuated or killed whole-cell (KWC) vaccines. Live attenuated vaccines have been extensively explored for mucosal immunization in humans and animals for decades. They are made from weakened versions of infectious pathogens, produced using physical, chemical, or biological methods, and contain antigens that closely mimic a true infection [188]. The most important advantages of live attenuated vaccines include long-lasting immunity, high efficacy, and non-parenteral administration. The manufacture and storage of those vaccines, however, can be more difficult than other types of vaccines. In addition, since the agent is living, there is a chance that the agent can revert to wild type or that secondary mutations may transform attenuated vaccine strains into more contagious and virulent forms [47,188,189].

Killed whole-cell (KWC) bacterial vaccines are produced from inactivated or killed microorganisms that are incapable of replicating within the hosts. KWC vaccines are typically more stable, requiring easier-to-maintain storage conditions [188]. They can also be lyophilized and stored at ambient temperature for long times without affecting stability and efficacy, like inactivated whole-cell-based cholera vaccines containing cholera toxin B subunit [190]. However, a replicating or quasi-replicating vaccine, like a viral-vectored vaccine, may elicit better immune responses, as initial increases in antigen exposure may be interpreted by the immune system as a signal of a threatening infection [191], although dosing schedules or dose delivery technologies to non-live or quasi-live vaccines may yield improve immune responses. Several studies have shown that limited antigen exposure can lead to poor immunogenicity, especially to highly conserved epitopes, which could potentially induce epitope-specific broadly immune responses but are mostly immune subdominant in nature, such as influenza hemagglutinin stalk domain or the fusion peptides from some envelope proteins of viruses with Type 1 viral fusion proteins [102,192,193,194,195]. To overcome this, an adjuvant can be included with a KWC vaccine to augment efficacy by stimulating immune responses or modulating the presentation of antigen, but with potential risks of increased reactogenicity and adverse reactions like inflammation [189]. Some non-replicating vaccines require booster doses and reimmunization [47,189]. However, the disadvantages seen with KWC vaccines exist for all non-replicating vaccines, including, for example, vaccines based on proteins and polysaccharides and mRNA vaccines. A significant advantage of KWC vaccine vector platforms and the other non-replicating vaccine platforms is that antivector immunity will not be a barrier to the use of the platform for multiple different vaccines. Antivector immunity can create significant problems for viral-vectored vaccines, for example, adenovirus or poxvirus-vectored vaccines [196,197,198], making it difficult to use the same virus vector for more than one disease antigen. For KWC vaccines, antivector immunity could even have a potential adjuvating effect.

A vaccine approach that can be particularly useful for mucosal vaccines is subunit vaccines, which are made from isolated components of microorganisms instead of whole organisms. Subunit vaccines are often easier to produce and store than traditional live attenuated vaccines. Subunit vaccines are the basis for many vaccine technologies, like isolated protein and polysaccharide and mRNA vaccines. Subunit vaccines have the advantage that they aim to elicit immune responses against the component of the pathogen that is known to be the specific target of immune responses that confers protective immunity against the pathogen, for example, a viral envelope protein involved in virion binding and entry into host cells, or microbial toxins that are responsible for the pathogenesis of disease. Another compelling advantage of subunit vaccines is that they can be designed to elicit immunity against a part of a pathogen antigen that is highly conserved across different pathogen variants, enabling a route to a variant-resistant vaccine or even a vaccine that provides protection against a large family of pathogens. A vaccine that can elicit protection against a highly conserved antigen may be particularly useful for pathogens that rapidly evolve variants within an infected individual, like HIV, or across a population, like coronaviruses. One licensed mucosal vaccine based on pentameric cholera toxin B subunit (CTB) effectively prevents disease. CTB has promoted the effective delivery of bound heterologous antigens to mucosal APCs, such as enterocytes, M cells, macrophages, and DCs [199,200,201,202]. Human clinical studies have demonstrated that CTB included in a vaccine preparation promotes the induction of broadly reactive antigen-specific local IgA and systemic IgG responses via rectal and intranasal administration [203,204,205]. Recent studies have also shown that CTB elicits mucosal T-cell immune responses [206,207], including local and intestinal protective tissue-resident memory T (T_RM_)-cells [47,208].

Consideration of the advantages and disadvantages of traditional vaccine formulations has led to the development of several next-generation mucosal vaccines. These include viral vector-based vaccines, particle-based vaccines, and vaccine platforms employing native or modified whole bacterial cells to deliver and present antigens. Viral vector-based vaccines use biomodified replicating or non-replicating viral vectors to deliver DNA or RNA instructions for antigen production to vaccine cells, enabling antigen production by the cells, followed by the induction of immune responses against the vaccine antigen expressed by the cells. Several viral vectors have been successfully developed as vaccine-delivery vectors. A widely used approach employs adenoviral vectors to deliver the genetic instructions. Adenoviral vector vaccines can elicit both strong humoral and cellular immune responses, even to the point of sterilizing immunity [209,210,211,212,213,214,215]. Viral vector-based vaccines have been shown to elicit good immunity against diseases that are transmitted across mucosal surfaces with ease of administration [216,217,218,219]. The formulation and delivery of viral vector-based vaccines can vary depending on the type of vaccine and the disease it is designed to prevent. One major hurdle of viral vector-based vaccine application is that previous exposure to the vector can reduce effectiveness, which prevents multiple uses of the same vector. Pre-existing immunity to microbial vectors can limit the vaccine’s effectiveness, even for priming with viral-vectored vaccines [47]. Although a recent human clinical study of a COVID-19 mucosal vaccine based on recombinant adenovirus serotype 5 (Ad5) vector has shown some efficacy, with safety and tolerability [220], there are still concerns about how vaccine efficacy may be affected by pre-existing antivector immunity, as observed in two HIV-1 vaccine human clinical studies, Step and HVTN 503/Phambili, more than a decade ago [221,222,223]. Both studies used recombinant adenovirus serotype 5-vectored vaccines and failed after phase 2b studies as it was found that populations who were Ad5 seropositive before vaccination had an increased risk of HIV-1 acquisition after vaccination [224,225]. Later, several follow-up studies explored the potential underlying mechanisms of how vector-specific pre-existing immunity correlates with risk of infection, including an association of high frequencies of preimmunization Ad5-specific T-cells with decreased magnitude of HIV-specific CD4 responses and decreased breadth of HIV-specific CD8 responses [226], enhancement of HIV-1 replication in CD4 T-cells by Ad5 immune complexes activation of the dendritic cell–T-cell axis [227], increased susceptibility to HIV infection by Ad5-specific CD4 T-cells [228], and studies on non-human primate challenge models further implicated increased the risk of SIV acquisition from low-dose SIV penile challenge associated with higher pre-immune responses to Ad5 vectors [229], suggesting that pre-existing Ad5 immunity might dampen desired vaccine-induced responses and increase the risks of viral acquisition of the HIV-1 infected population.

Particle-based vaccines are relatively new types of vaccines. These include nanoparticle, microparticle-based vaccines, and virus-like particles (VLPs). Particle vaccines serve not only to deliver the antigen cargo to the host but also to modulate antigen release rates since prolonged antigen release may enhance the level and quality of immune responses [230,231,232]. Some KWC microbial vectors that express vaccine antigens can be considered biologically produced microparticles. Other advantages of particle vaccines include (1) protection of the antigens; (2) due to the size of the particles, they can facilitate capture and uptake by APCs followed by maturation and stimulation of those APCs to further activate innate and adaptive immunities; (3) particle vaccines are versatile platforms to delivery several different antigens or combination of antigen with immune stimulatory molecules. Some particles can also act as adjuvants [232,233]. Indeed, some of the impressive effectiveness of the recently developed mRNA vaccines may be attributed to the adjuvating effects of the lipid nanoparticles used to encapsulate the mRNA component of the vaccines [234]. Several different chemical or biomaterials have been utilized to develop mucosal particle-based vaccines such as chitosan [235,236], poly lactic-co-glycolic acid (PLGA) [236,237], recombinant self-assemble protein subunits [238,239,240], lipid bilayers (FDA approved mRNA COVID-19 vaccines) [241,242], and VLPs that mimic viruses but without infectious components [243,244,245]. Those particle-based mucosal vaccines, either administrated alone or in combination with adjuvants, have been proven to be good at stimulating the immune system and eliciting effective immune protection against infections [230,232,244,246]. Particle-based vaccines beyond the licensed mRNA–lipid nanoparticle vaccines are still in the early stages of development, and more research is needed to determine their effectiveness and safety. However, they have the potential to be a promising new approach to vaccination.

Benefiting from the fast development of modern molecular biotechnologies, traditional whole-cell antigen platforms have the potential for combination with purified recombinant subunits but also for overexpression of desired antigens on surfaces of those whole-cell vectors [47]. Results from several groups attempting to use whole bacterial cells engineered to overexpress antigens on the cell surface suggest that expression of antigens on killed or inactivated whole bacteria cell surfaces offers a promising approach to elicit effective immune protection. The killed bacteria cells not only deliver the antigens but also act as adjuvants and help minimize the required vaccine doses [20,102,247,248]. Moreover, those killed whole-cell vaccines are easily produced by industry on a large scale at a low cost, are stable at refrigerator temperatures for prolonged storage, and are easy to administer through mucosal immunizations [247,248]. All these characteristics make the killed whole-cell vaccine a potentially affordable vaccine candidate, especially for lower-income countries, to help reduce the burden of infectious diseases, particularly in response to pandemics [63].

Mucosal vaccine delivery strategies include administration across various mucosal surfaces like the nose, mouth, and lungs. Compared with traditional injection-based vaccination, mucosal vaccine delivery methods can stimulate both mucosal and systemic immunity to provide not only quicker but also better protection against infections, given that infection most frequently happens across mucosal surfaces. Researchers have also found that some mucosal vaccines can induce strong resident memory T-cell responses in mucosal-associated lymphoid tissues to provide durable protective immunity [249]. The delivery of mucosal vaccines can also vary depending on the type of vaccine and the disease they are designed to prevent. Several strategies have been used to deliver mucosal vaccines, including oral delivery by pill, liquid, dissolvable substrate, or small patches. Intranasal and respiratory delivery mechanisms have included nebulizers, nasal sprays, and inhalers. There is an approved nasal vaccine to prevent influenza and an oral live-attenuated bacterial vaccine, TY21A (Vivotif^®^), against *Salmonella* to prevent typhoid fever [43]. The vaccine delivery method, along with factors such as vaccine formulation, target antigen/pathogen, and host immune status, will significantly affect the development of protective immunity.

## 6. Mucosal Adjuvants

Many challenges hamper the efficacy of mucosal vaccines. One approach to improving mucosal vaccine efficacy involves the use of adjuvants. Mucosal vaccine adjuvants are substances and vaccine components that enhance immune responses to antigens delivered through mucosal surfaces [103,104]. Adjuvants act by various mechanisms, such as increasing antigen stability, optimizing antigen delivery and exposure pharmacokinetics, activating innate immune receptors, modulating mucosal barriers and secretions, recruiting and activating immune cells, and stimulating mucosal cellular immune responses [250,251,252,253]. Mucosal vaccine adjuvants can also allow for a reduced antigen dose or enable good immune responses to vaccines with fewer doses. Mucosal vaccine adjuvants that have been studied include bacterial toxins and their derivatives, CpG-containing DNA, cytokines and chemokines, nanoparticles, and biopolymers (Table 2) [254].

Bacterial enterotoxins and their derivatives have been studied and used for mucosal vaccine adjuvants, including cholera toxin (CT) and heat-labile enterotoxin from *E. coli* (LT) [279]. These adjuvants induce high IgA antibody levels and long-lasting memory to target antigens when administrated mucosally [280]. The mechanisms underlining the enhanced immunogenicity have also been investigated extensively. Recent studies have demonstrated that bacterial enterotoxin adjuvants promote DC cell migration from the subepithelial dome to the follicular-associated epithelium after vaccine administration, followed by antigen capture and presentation in mesenteric lymph nodes [104,280,281,282]. These results showed the critical role of APC stimulation by adjuvants plays in inducing both local mucosal responses and mucosal surface crosstalk [282]. Enterotoxin adjuvants can also induce mixed Th1/Th2 immune responses [280], and the adjuvants can promote Th17 responses that are correlated with high levels of IgA antibody, indicating the importance of IL-17, expressed by Th17 cells, in promoting vaccine-induced protection [283,284]. Although bacterial enterotoxins and their derivatives serve as effective mucosal adjuvants, toxicity concerns have prevented widespread use in clinical applications [285]. However, some toxin subunits or otherwise modified or mutated toxins are used in approved vaccines, such as the recombinant cholera toxin subunit B used in Dukoral^®^ [41].

Another class of adjuvants include pattern recognition receptor (PRR) ligands. These adjuvants yield enhanced immune responses by toll-like receptor (TLR)-mediated immune cell activation through pathogen-associated molecular pattern (PAMP) recognition. Synthetic oligodeoxynucleotides (ODNs) containing unmethylated CpG is a TLR9 agonist [286]. It activates the innate immune responses to yield strong Th1 responses and proinflammatory cytokines, inducing cytotoxic T lymphocyte (CTL) responses and interferon (INF)-γ secretion [287]. CpG also activates antigen-presenting cells (APCs) by upregulating the expression of MHC, CD40, and CD80 on DCs to enhance antigen processing and presentation [287]. CpG ODNs have been found to stimulate TLR9-expressing B lymphocytes, leading to increased IgA levels [288]. In 2017, the US FDA approved HEPLISAV-B^®^ for hepatitis B, the first vaccine to employ a CpG ODN adjuvant [289]. Other nucleotide-based adjuvants include cyclic dinucleotides, such as c-di-GMP and c-di-AMP, which are ligands for the stimulator of interferon genes (STING) pathway that can induce type I interferon and proinflammatory cytokines to enhance antigen presentation as well as adaptive immune responses [290,291,292]. Cyclic dinucleotides improve mucosal immunity against influenza, tuberculosis, and anthrax [254].

Monophosphoryl lipid A (MPL), derived from lipopolysaccharide (LPS) of Gram-negative bacteria, is another promising adjuvant that stimulates immune responses by activating TLR4 [293]. When used in complex emulsion liposome formulations, plus the active fraction of the bark of Quillaja saponaria (QS-21) or alum, MPL induces strong, antigen-specific CD4+ T-cell responses [294,295,296]. Flagellin, a key component of bacterial flagella, is a natural agonist of TLR5 and another mucosal adjuvant. After binding to TLR5, flagellin activates the MyD88-dependent pathway, which mobilizes nuclear factor NF-κB and stimulates tumor necrosis factor (TNF)-α production [297,298]. Flagellin is also recognized by two NOD-like receptors (NLD), NLRC4 and NAIP5, to activate the inflammasome response [103,299]. Flagellin is a potent adjuvant that can stimulate both innate and adaptive immune responses as it activates proinflammatory cytokine and chemokine production in various immune cells in the mucosa, including DCs, natural killer cells (NKs), epithelial cells, and lymph node stromal cells [103,300].

Cytokines and chemokines, including IL-6, IL-12, and IL-15, help induce CTL responses and antigen-specific IgA antibodies at mucosal sites, while cytokines from the IL-1 family induce both IgA and IgG [256,301]. Other cytokines and chemokines, such as IFN-γ, IL-2, IL-18, IL-21, and granulocyte-macrophage colony-stimulating factor (GM-CSF), exhibit adjuvant activity at mucosal sites [103,302]. However, dose-related toxicities of those cytokines and chemokines, as well as concerns about the stabilities and efficient delivery methods, hinder their clinical application [302,303].

Several natural-product-based mucosal vaccine adjuvant candidates have been studied and may have advantages over conventional adjuvants [304] as they promote specific and non-specific immunity in the host with lower toxicity and side effects due to their biocompatibility and biodegradability [304,305,306]. These adjuvants include biopolymers, nanoparticles, virus-like particles, and extracellular vesicles [304]. N-dihydrogalactochitosan (GC), a novel biopolymer synthesized from galactose and chitosan, has mucoadhesive, pH-resistant, and biocompatible properties. GC can induce type I interferon production and enhance humoral and cellular immunity against SARS-CoV-2. Nanoparticles, such as ferritin and virus-like particles (VLPs), can encapsulate or display antigens in combination with stimulatory molecules on their surfaces with adjuvant activities. The nanoparticles can protect antigens and adjuvant compounds from degradation, increase antigen uptake by mucosal cells, and target specific immune cells and tissues to further enhance mucosal immunity [304].

Mucosal vaccine adjuvants are an active area of research, as mucosal vaccines made using the adjuvants offer many advantages over conventional parenteral vaccines. However, mucosal vaccine adjuvants also face challenges, such as safety, toxicity, stability, and regulatory approval [103,104]. The development of mucosal vaccine adjuvants is at an early stage, and more research is required before they become widely employed in licensed vaccines.

## 7. Challenges in Mucosal Vaccine Development

Mucosal vaccines promise to elicit protective immunity against infection with substantial advantages over traditional parenteral vaccines. While the development of new mucosal vaccines is highly desirable, more work is needed as there are still challenges in this field. Identification of the correlates of protection is one of the major concerns. This includes a better understanding of mucosal immunity and the key components that contribute to durable and broadly effective protection at mucosal sites after natural infections and vaccinations. Also needed is an improved understanding of the underlying molecular and cellular mechanisms responsible for optimal mucosal immunity, which will facilitate preclinical research, clinical development, and regulatory approval. Important efforts would include basic research aimed at revealing the specific immune cells critical for protective mucosal immunity and the underlying molecular mechanisms that lead to the induction of immunity at mucosal surfaces [47,307,308,309]. Improved understanding is also needed of the host mucosal microenvironment, including the mucosal microbiome, that can affect biophysiological features of the mucosal barrier to help regulate immune responses [310,311,312].

Improved animal models will also be needed to further the development of mucosal vaccines. Existing, widely used animal models, particularly small animal models, may not be adequate and may not correctly predict the activity of mucosal vaccines in humans or other larger animals, such as agriculturally important animals. Beyond simple differences in the sizes of the organs and systems with mucosal surfaces and the fact that several commonly used small animal models are not native hosts for diseases that would be the targets for mucosal vaccine development, the mucosa structures and those associated immune systems in animal gastrointestinal, respiratory, and urogenital tracts vary from those in humans and other larger animals [45]. For example, although mouse models have been widely used for influenza vaccine development, there are limitations since the species is not the natural host, and the sialic acid receptors that mediate viral entry and their distribution in the respiratory tract are different from those in humans. Ferrets are considered more suitable models since they are more biologically similar to human beings in being able to be infected with human influenza virus isolates, exhibit pathogenicity that is closer to that experienced by humans, have more homologous sialic acid receptor distribution, can be infected via airborne transmission, and will mount protective immune responses after vaccination [313,314]. Besides the natural features of the model species, the immune responses elicited by vaccination and by pathogen challenge also need to be validated in comparison to human natural infection or vaccination [313,314,315]. In general, more predictive animal models are needed to help develop new safe and effective mucosal vaccines for a large variety of pathogens.

Determining the ideal vaccine formulation and delivery system is another major hurdle for mucosal vaccine development. Efforts include defining vaccine formulations based on the type of pathogen and the desired immune responses, especially immune responses that correlate well with protection [42,45,47,104]. Determination of dosage and timing are also important [45,47]. Finding better, safe, and effective adjuvants is another critical aspect [103,104]. The methods and devices for vaccine administration are additional areas needing development. Factors such as manufacturing, supply chain, stability, acceptability, and cost are key practical considerations for mucosal vaccine development [45]. Other challenges for mucosal vaccine development include conducting clinical trials that are both acceptable to regulatory authorities internationally and appropriate for at-risk patient populations [45,47].

Mucosal vaccination, beyond the currently approved vaccines, holds great promise for the future. While substantial challenges still hamper the development of safe and effective mucosal vaccines, researchers are continuing to create exciting new technologies to enable the development of future mucosal vaccines.

## 8. Disadvantages and Side Effects of Mucosal Vaccines

Currently, licensed mucosal vaccines can produce adverse effects. The live-attenuated influenza intranasal vaccine has caused some rare incidences of asthma exacerbation [316]. This has been a concern for parenterally administered live attenuated influenza vaccines as well, but they are now generally considered safe for patients with asthma [317]. More common adverse effects for some of the oral vaccines include fatigue, headaches, and gastrointestinal symptoms [318,319], but such adverse effects are common for many vaccines or even many therapeutics. Some rare cases of anaphylaxis and Guillain–Barré syndrome have also been reported for adenovirus/adenovirus-vectored vaccine [320]. None of these side effects are limited to mucosal vaccines, and similar adverse effects have been caused by various parenteral vaccines [321,322].

Live attenuated vaccines, which constitute the most common mucosal vaccines, come with an inherent risk of reversion to the pathogenic wild type [16]. Vaccine-associated poliomyelitis is contracted at a rate of 4.7 cases per million births by both vaccinees and their contacts [323].

Intranasal vaccines, in particular, must be rigorously tested for safety since the proximity of the nasal passages to the olfactory bulbs may allow access to the central nervous system. This is more of a concern for living or quasi-living (viral-vectored) intranasal vaccines. One study showed that adenovirus vectors are able to infect the CNS when administered intranasally [324]. There are concerns that live attenuated intranasal vaccines, particularly influenza, could cause encephalitis [28]. However, this has not been proven to be a problem [316].

Inactivated or subunit mucosal vaccines may be safer than live vaccines, but they usually require the addition of an adjuvant [324]. These adjuvants, especially enterotoxins, have safety concerns [104]. For example, a cholera toxin adjuvant in an inactivated influenza vaccine increased the risk of Bell’s palsy by almost twenty times when the vaccine was administered intranasally [325,326]. For oral vaccines, it remains a challenge to find adjuvants that are potent enough to stimulate an immune response and survive passage through the upper gastrointestinal tract while minimizing gastrointestinal symptoms [327].

One of the biggest disadvantages of oral vaccine development is the potential for tolerance to the antigen. Chronic environmental enteropathy and deficiencies in vitamin A or zinc, which are more likely to occur in developing countries, significantly decrease the immune response to oral vaccines [52,328]. Additionally, repeated doses of an antigen given orally can result in immune tolerance [329,330,331]. This must be taken into account when considering the number of boosts an oral vaccine needs in order to avoid promoting tolerance to the pathogen.

## 9. Conclusions and Perspectives

Mucosal vaccines hold great promise to prevent infection and transmission of pathogens, most of which enter hosts across mucosal surfaces. Mucosal vaccines can elicit local immunity at mucosal sites while also inducing strong and long-lasting systemic immune responses. Many mucosal vaccines for humans and animals have been developed and proved effective in blocking infection and transmission of pathogens. Nevertheless, there are still drawbacks to current mucosal vaccines and challenges for new mucosal vaccine development. The importance of rapid production of effective, safe, cost-efficient, distribution-friendly, easily administered mucosal vaccines manufactured at large scale in locations critically affected by disease has become increasingly obvious, particularly given recent experience with COVID-19. Dedicated efforts to advance the mucosal vaccine field and lower the barrier of mucosal vaccine development are essential to address current societal needs and to prepare for the next potential pandemic.

## Figures and Tables

**Figure 1 vaccines-12-00191-f001:**
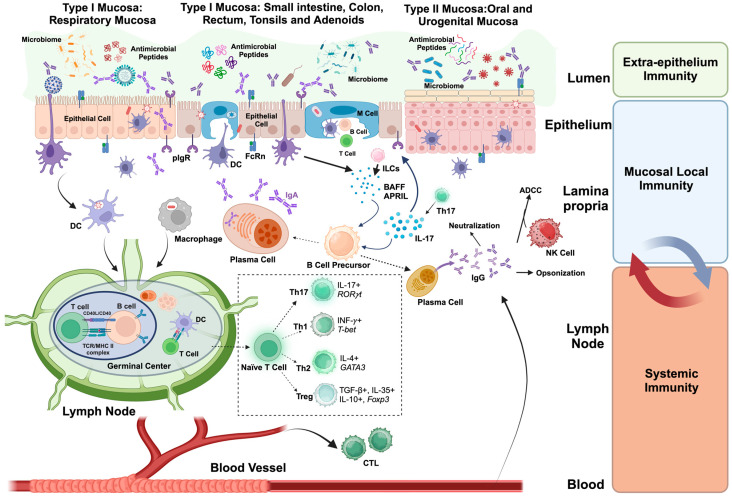
Mucosa structures and immunity mechanisms. Mucosal immune responses are composed of multiple immune effector mechanisms with a complex network of innate and adaptive immune components and play critical roles in defending against invasive pathogens at host mucosal barriers. The first defense line of mucosal barriers is epithelial cells, which have tight cell junctions serving as physical barriers, together with microbiota and antimicrobial peptides in the lumen as biological and biochemical barriers, respectively. Ciliated cells in the respiratory tract epithelium are important for mucociliary clearance by propelling pathogens and particles out of the airways. Mucosal epithelium can be defined into different types: nonkeratinized epithelium, columnar epithelium, and squamous epithelium (the squamous epithelial cells in the respiratory tract and ends of the GI tract are omitted from the figure) according to their structures and functions. Pathogen surveillance is mediated by M cells (antigen transport) and DCs (antigen processing and transport) at mucosal barriers. Antigens are captured and processed by DCs and macrophages, leading to the activation and maturation of those antigen-presenting cells (APCs). Antigen-stimulated DCs later initiate downstream immune responses by migrating to lymph nodes to present antigens on MHC molecules to T-cells, which further mediate expansion of naïve T-cells into differentiated T helper subsets (Th1, Th2, Th17, Treg) via regulation of transcription factors and secretion of lineage-defining cytokines. Activated T helper subsets then perform their functions, including upregulation of polymeric Ig receptor (pIgR) expression and promotion of B-cell differentiation into plasma cells, with IgA class switching to produce secretory IgA (SIgA) to intercept the pathogens at the mucosa. Soluble factors (BAFF, APRIL) secreted by DCs, innate lymphoid cells (ILCs), and epithelial cells could also enhance IgA production by T-cell independent class switching. IgG, derived from local B-cells or from blood, is also present in mucosal tissues to directly neutralize pathogens or mediate cytotoxicity by recognition of Fc receptor expressed on nature killer (NK) cells. In addition, antigen-specific cytotoxic T-cells (CTLs) also enter the mucosa to kill those infected cells. (Figure created with BioRender.com).

**Figure 2 vaccines-12-00191-f002:**
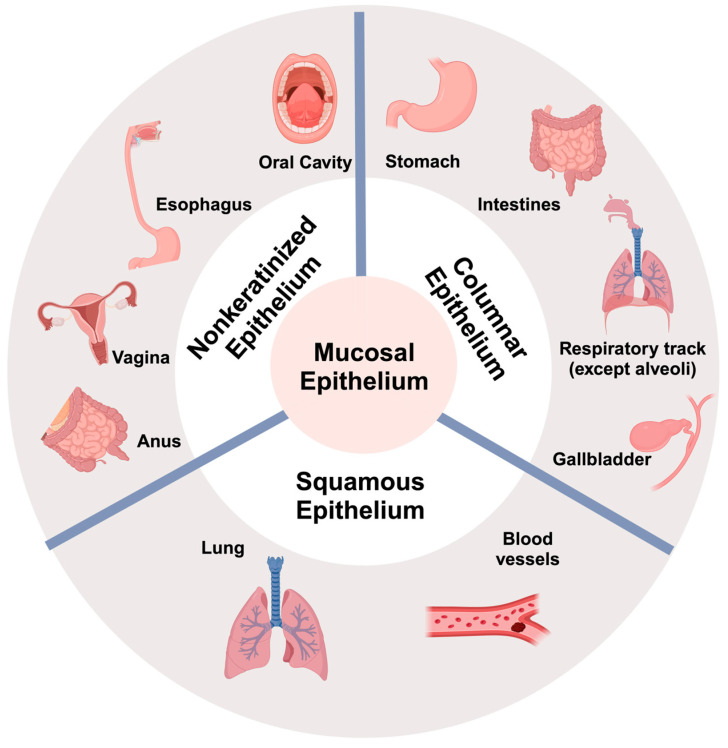
Different types of mucosal epithelium structures and functions. The oral cavity, esophagus, vagina, and anus have nonkeratinized epithelium composed of several layers of cells. The columnar epithelium is found in the stomach, intestines, and respiratory tract, except for the alveoli and gallbladder, which are composed of tall, narrow cells that are tightly packed together. Squamous epithelium is a type of epithelium found in the lungs, blood vessels, and body cavities, which are composed of flat, scale-like cells that are tightly packed together. (Figure created with BioRender.com).

**Table 2 vaccines-12-00191-t002:** List of vaccine adjuvants licensed or in clinical trials.

Adjuvant Name	Type/Composition	Immune Stimulation Function	Development Stages	Disease Type	References
Cholera Toxin (CT) *	Recombinant detoxified bacterial A–B subunit	Gangliosides binding; cAMP stimulation; DC activation; mucosal IgA and Th production	licensed for human use (Dukoral^®^)	*Vibrio cholerae*	[41,103,255,256]
Alum	Aluminum mineral salt; Insoluble particulates of hydroxide, phosphate, or hydroxyphosphate sulfate salts	Antigen adsorption and controlled release; Humoral immunity modulations; Induction of Th2 immune responses and inflammation	licensed for human use (Daptacel^®^, Twinrix^®^, Gardasil^®^, Bexsero^®^, Prevnar 20^®^)	HAV, HBV, HPV, Diphtheria, Tetanus, *Haemophilus influenza*, Meningococcal, Pneumococcal	[254,255]
MF59, AS03 *	Oil-in-water (squalene in water) emulsion	APC stimulation and activation; Modulation of humoral and cellular immune responses	licensed for human use (Fluad^®^, Pandemrix^®^, Arepanrix™)	Influenza (pandemic and seasonal)	[254,255,257,258,259]
CpG ODN (e.g., 1018 ISS)	Oligonucleotide	Soluble TLR9 ligand; Increase both humoral and cellular immune responses; Th1-specific cytokine expression and CD8+ T-cell activity	licensed for human use (Heplisav-B™, CorbeVax™)	HBV, SARS-CoV-2	[103,254,258]
MPL *, AS04 *	Non-toxic derivative of LPS (MPL); Alum-adsorbed MPL (AS04)	TLR4 agonist; Increase APC maturation; Induction of Th1 type immunity and improve both humoral and cellular immune responses	licensed for human use (Cervarix™, Fendrix™)	HBV, HPV	[103,254,255,258,259]
VLP *, Virosomes *, Liposomes (IRIV) *	Liposomes, viral proteins, and nanoparticles	PAMP signals; Improved APC antigen uptake to promote T-cell and B-cell activation	licensed for human use (Epaxal^®^, Engerix^®^-B, Gardasil^®^, Cervarix™)	HBV, HPV, HAV, Influenza	[254,255,257,259]
Alhydroxiquim-II	Alum adsorbed TLR7/8 agonist	Stimulate TLR7 and TLR8; Lymph node-specific delivery and antigen release	licensed for human use (COVAXIN^®^)	SARS-CoV-2	[254]
QS-21	Purified plant extract containing water-soluble triterpene glycosides (saponins)	Activation of strong humoral and cellular immune responses; stimulation of cytotoxic T-cell response	licensed for human use (Shingrix^®^, Arexvy^®^, Mosquirix^®^)	Varicella-zoster virus, RSV, Malaria	[260,261]
AS01 *	Liposomes containing MPLA and saponin QS-21	TLR4 ligand; Induction of Th1 type and CD8+ T-cell mediated immunity; Augment of antibody responses	Phase III	Malaria, (RTS,S) and herpes zoster vaccine (HZ/su)	[262]
EGVac system	Bacterial polysaccharide and DNA	Stimulation of both B-cell and T-cell immune responses	Phase II	HPV	[258,263]
Saponin complexes *, ISCOM *, QS-21 *, Matrix-M *	Nanocomplex of lipid, pure saponins, and cholesterol	Induction of humoral immunity; Th1, Th2 and CD8+ T-cell responses	Phase I	Influenza	[258,259,264,265,266]
ISA51	Oil-in-water emulsion stabilized with non-ionic surfactant	Augments antibody responses and T-cell activity	Phase II	Influenza (seasonal)	[267,268]
VAX2012Q, VAX125	TLR5 agonist, bacterial flagellin fused antigen	Promotion of antibody titer and T-cell immunity	Phase II	Influenza	[258,269]
Ampligen^®^ *, rintatolimod, PIKA	Poly I:C, double-stranded (ds)-RNA polymer analog, TLR3 agonist	Enhanced humoral responses and Th1; Th17 responses	Phase I and Phase III	Influenza, Rabies, HIV-1	[254,270,271,272]
Matrix-M *	Protein nanoparticle	Improved antibody responses and T-cell activities	Phase II	SARS-CoV-2	[273]
GM-CSF *	Granulocyte-macrophage colony-stimulating factor, cytokine	Enhanced mucosal IgA responses; activation of DCs and T-cells	Phase I and Phase II	HIV-1, SARS-CoV-2	[274,275]
IFN-α *	Type 1 interferon (IFN)	Induction of secretory IgA (sIgA) response in saliva	Phase I	Influenza	[276]
CCL3 *	Chemokine	Secretion of mucosal IgA and CTL	Pre-clinical	HIV-1	[270]
α-GalCer *	Glycosphingolipids	Activation of pattern recognition receptor; induction of IgA responses and Th1/CTL responses	Phase II	HBV	[277]
Chitosan *	Natural-product-based particulate-polysaccharides,	Enhanced mucosal T-cell responses; induction of proinflammatory cytokines and secretory IgA responses and Th2 responses; PRR activation; upregulation of co-stimulatory molecules; activation of complement pathways	Phase II	Norovirus	[278]

*: Adjuvants developed for mucosal vaccines.

## Data Availability

Figure 1 and Figure 2 are created with BioRender.com.

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
