# Peer review of "Vaccine Strategies to Elicit Mucosal Immunity"

_vaccines, 2024, doi:10.3390/vaccines12020191_

Round 1

Reviewer 1 Report

Comments and Suggestions for Authors

This is a well written review that captures a great amount of information on the top of mucosal vaccines and adjuvants.  I have a few comments for the authors:

·         Although there is a good use of references, the authors gave more emphasis on enteric vaccines, whereas nasal vaccines against res pathogens have been underrepresented. For instance, recently published studies on nasal vaccines against pertussis and TB.

·         On mucosal immunity section:  In the human airways, IgA is the predominant immunoglobulin, however in the lung mucosa IgG in substantially increased and often measured in higher levels comparted to IgA.

·         Less emphasis has been given on resident T cell reservoirs, and how their long-lived properties and cross-reactive potentials of TRM can be key for broader and longer immunity at mucosal sites.

·         Overall, the narrative flows well, but there are quite a few repetitions throughout the review (sometimes whole sentences). I would suggest that the authors try to remove repetitive information and sharpen some sentences.

Author Response

Thank you for the very helpful reviews. We have worked to respond to the comments and believe that the manuscript is now significantly improved. We outline our responses to the reviewers’ comments point-by-point below.

Please note that the line numbers cited below refer to the lines in the clean revised version with referred modification highlighted.

Reviewer 1:

This is a well written review that captures a great amount of information on the top of mucosal vaccines and adjuvants.

Thank you for the kind remarks.

I have a few comments for the authors:

Although there is a good use of references, the authors gave more emphasis on enteric vaccines, whereas nasal vaccines against res pathogens have been underrepresented. For instance, recently published studies on nasal vaccines against pertussis and TB.

Responses:

We added information regarding developments in intranasal vaccines for pertussis, TB, and SARS-CoV-2. That new information was added from line 164-181 [refs 85-101] in in the clean revised version with referred modification highlighted.

It was also added to lines 139-140 [refs 69-74].

On mucosal immunity section: In the human airways, IgA is the predominant immunoglobulin, however in the lung mucosa IgG in substantially increased and often measured in higher levels comparted to IgA.

Responses:

We modified information from line 425-431 [refs 154,156,175-177] in the clean revised version with referred modification highlighted.

Less emphasis has been given on resident T cell reservoirs, and how their long-lived properties and cross-reactive potentials of TRM can be key for broader and longer immunity at mucosal sites.

Responses:

We added more information on resident T cell reservoirs from line 469-477 [refs 61,182] in the clean revised version with referred modification highlighted.

Overall, the narrative flows well, but there are quite a few repetitions throughout the review (sometimes whole sentences). I would suggest that the authors try to remove repetitive information and sharpen some sentences.

Responses:

We have gone through the paper and tried to identify and consolidate repetitive information. We believe that the manuscript now has less repetition and a sharper focus.

Reviewer 2 Report

Comments and Suggestions for Authors

This literature review paper talks about mucosal vaccines from the aspect of history, basic biology, and vaccine development. Overall, the paper is organized and well written with a lot of knowledge. Readers from different fields will be benefited from reading this paper. Just a few comments:

1.     Some lines need references, such as 97-99, 181-188, just for instance. Usually, we cite papers after a solid statement. Please go over the entire manuscript and make sure papers are well cited. 

2.     Please re-organize figure one. Currently, things look a little messy, and the font size in the main figure is small. It will also be nice to keep the font size consistent. 

3.     You mentioned the word safe for multiple times, however, it would be nice to discuss more about it to make things more comprehensive. For instance, is there any known side effects or disadvantage? 

Author Response

Thank you for the very helpful reviews. We have worked to respond to the comments and believe that the manuscript is now significantly improved. We outline our responses to the reviewers’ comments point-by-point below.

Please note that the line numbers cited below refer to the lines in the clean revised version with referred modification highlighted.

Reviewer 2:

This literature review paper talks about mucosal vaccines from the aspect of history, basic biology, and vaccine development. Overall, the paper is organized and well written with a lot of knowledge. Readers from different fields will be benefited from reading this paper.

Thank you for the kind remarks.

Just a few comments:

  1. Some lines need references, such as 97-99, 181-188,
    just for instance. Usually, we cite papers after a solid statement. Please go over the entire manuscript and make sure papers are well cited.

Thank you for this suggestion. We have added a number of references including the following in the clean revised version with referred modification highlighted.

Lines 99: added reference [49,50]

Lines 64-68: added reference [28]

Lines 122-123: added reference [57,59]

  1. Please re-organize figure one. Currently, things look a little messy, and the font size in the main figure is small. It will also be nice to keep the font size consistent.

We have revised Figure 1 in response to the suggestions. We believe that its appearance has now been improved.

  1. You mentioned the word safe for multiple times, however, it would be nice to discuss more about it to make things more comprehensive. For instance, is there any known side effects or disadvantage?

We added new material in Section 8 from line 813-849 [refs 16, 28, 52, 104, 317-332] in the clean revised version with referred modification highlighted.

Reviewer 3 Report

Comments and Suggestions for Authors

This review by Song and colleagues represents and interesting piece of data regarding the current knowledge of mucosal vaccinology. It is like a textbook, not going into too much detail, but gathering most of the relevant information you would like to know when approaching this field. I am including some comments to improve its impact and to broaden to some aspects that I think is worth to explore.

1.       Abstract seems an introduction more than a summary of the review content. Please, reword the abstract in order to provide an idea of what the reader will find in this manuscript.

2.       Lines 251 – 263 represents a compilation of different types of epithelial cells with no references supporting the text. I would recommend using rather a table to show these data, adding the specific reference where the description of each cell type is coming from. This could even appy to text in lines 280 – 288.

3.       Lines 306 – 321. Mucosal cells can also produce cytokines. This should be commented and quoted in this section.

4.       Section between lines 344 and 369. The existence and function of T resident cells (Trm) should be discussed in this section as they appear afterwards in the manuscript (line 525).

5.       Sometimes, there is repetitive, too general information along the text. For instance, lines 435 to 444 explain the general antigen presentation process after discussing to some degree of detail the activation of T and B cells. This text could be introductory to the section 4 of the review, but places where it is, does not provide any additional information. Same could be said for lines 451 – 453.

6.       In my opinion, the review misses some mechanistic insights regarding how systemic responses are triggered following mucosal vaccination. This should be added.

7.       A section describing the disadvantages of mucosal versus parenteral vaccines should be included to make more reliable the review. Not everything is good in one side and bad in the other.

8.       In the last time, vaccines based on innate trained immunity (TI) are raising interest. It would be interesting to have in this nice review a perspective of this notion from the mucosal point of view. Can mucosal vaccination trigger TI? How does it work?

9.       Please, check the wording “is effectively prevents disease” in line 519.

Comments on the Quality of English Language

Sometimes the text is a bit repetitive. I would try to pulish this.

Author Response

Thank you for the very helpful reviews. We have worked to respond to the comments and believe that the manuscript is now significantly improved. We outline our responses to the reviewers’ comments point-by-point below.

Please note that the line numbers cited below refer to the lines in the clean revised version with referred modification highlighted.

Reviewer 3:

This review by Song and colleagues represents an interesting piece of data regarding the current knowledge of mucosal vaccinology. It is like a textbook, not going into too much detail, but gathering most of the relevant information you would like to know when approaching this field. I am including some comments to improve its impact and to broaden to some aspects that I think is worth to explore.

Thank you for the helpful comments and suggestions.

  1. Abstract seems an introduction more than a summary of the review content. Please, reword the abstract in order to provide an idea of what the reader will find in this manuscript.

We have substantially rewritten the abstract which is highlighted in the clean revised version with referred modification highlighted. We believe that the abstract now provides a better preview of the material in the manuscript.

  1. Lines 251 – 263 represents a compilation of different types of epithelial cells with no references supporting the text. I would recommend using rather a table to show these data, adding the specific reference where the description of each cell type is coming from. This could even apply to text in lines 280 – 288.

We have provided additional references from line 223-235 [refs 106-108, 111, 112] in the clean revised version with referred modification highlighted.

  1. Lines 306 – 321. Mucosal cells can also produce cytokines. This should be commented and quoted in this section.

Thank you for pointing this out. We added more material on this from line 367-370 [refs 149, 150] in the clean revised version with referred modification highlighted.

  1. Section between lines 344 and 369. The existence and function of T resident cells (Trm) should be discussed in this section as they appear afterwards in the manuscript (line 525).

Thank you for noticing this. We added more information on resident T cell reservoirs from line 469-477 [refs 61,182] in the clean revised version with referred modification highlighted.

  1. Sometimes, there is repetitive, too general information along the text. For instance, lines 435 to 444 explain the general antigen presentation process after discussing to some degree of detail the activation of T and B cells. This text could be introductory to the section 4 of the review, but places where it is, does not provide any additional information. Same could be said for lines 451 – 453.

Thank you for this helpful comment. We have gone through the manuscript and edited it in an effort to reduce repetitive information. The two sections mentioned above have been consolidated combined at the beginning of section 4.

  1. In my opinion, the review misses some mechanistic insights regarding how systemic responses are triggered following mucosal vaccination. This should be added.

Thank you for requesting this important information. We have worked to add more mechanistic information from line 478-490 [refs 42,47] in the clean revised version with referred modification highlighted.

  1. A section describing the disadvantages of mucosal versus parenteral vaccines should be included to make more reliable the review. Not everything is good in one side and bad in the other.

Thank you for pointing this out. We added new material in Section 8 from line 813-849 [refs 16, 28, 52, 104, 317-332] in the clean revised version with referred modification highlighted.

  1. In the last time, vaccines based on innate trained immunity (TI) are raising interest. It would be interesting to have in this nice review a perspective of this notion from the mucosal point of view. Can mucosal vaccination trigger TI? How does it work?

Thanks for this suggestion. We now have added more information on trained immunity from line 491-512 [refs 101, 183-186] in the clean revised version with referred modification highlighted.

  1. Please, check the wording “is effectively prevents disease” in line 519.

Thank you for catching this typo. We have now corrected it which is now line 576 in the clean revised version with referred modification highlighted.